

# Intent-aware knowledge graph-based model for electrical power material recommendation

Lin Zhao[1], Ning Luan[1], Weihua Cheng[1], Shuming Feng[1], Hui Wang[1], Yongcheng Yang[1] and Guixiang Zhu[2]

[1] Jiangsu Electric Power Information Technology Co. Ltd, Nanjing, China
[2] Nanjing University of Finance and Economics, Nanjing, China

## ABSTRACT

In the field of electrical power material management, it is paramount that users receive accurate recommendations regarding the electrical power materials they require. Recently, a growing number of studies have been dedicated to graph neural network (GNN)-based recommendation systems due to their ability to seamlessly combine node information with topological structure, enhancing the effectiveness of recommendations. However, a notable drawback of current GNN-based recommendation is their inability to explicitly capture users' intent in recommendations, which limits the performance. In fact, users' intent is crucial in determining their actions. One example is when users first form an intent to buy a particular set of items and then choose a specific item from the set based on their preferences. To fill this gap, this article proposes an intent-aware knowledge graph-based model for electrical material recommendation, named IKG-EMR. IKG-EMR models user preferences and intent by leveraging knowledge graph and user behavior sequences, respectively. Specifically, a graph neural network is adopted to generate user intent embedding and item embedding from the tripartite graph of "User-Item-Topic", and a multi-head attention network (Transformer) is used for extracting preference from user behavior sequences. Finally, an adaptive fusion with attention network is devised to generate comprehensive user representation by integrating user preference and intent features. Extensive experiments conducted on the real-life electric power materials show that our proposed model outperforms state-of-the-art methods.

## INTRODUCTION

With the ongoing digital transformation in the power industry, electrical power material management is encountering a range of issues. These include data silos, inefficiencies in information retrieval, and inaccuracies in forecasting material demands. These obstacles not only impede management effectiveness but also present risks such as supply chain interruptions and inventory imbalances, which can result in higher operational expenses (*You et al., 2023*). Therefore, there is a pressing requirement to address these challenges and elevate the sophistication of material management within the power sector.

Corresponding author
Guixiang Zhu,
zgx881205@gmail.com

In this context, leveraging theories such as big data and machine learning to predict the demand for power grid materials has emerged as a crucial strategy for modernizing material management practices. Recommendation systems, as sophisticated filtering tools powered by big data and artificial intelligence, have demonstrated considerable potential for application across diverse sectors, including e-commerce platforms, music streaming services, and news platforms. By analyzing users' historical behaviors and preferences effectively, recommendation systems can deliver personalized services, enhance user experience, and elevate satisfaction levels. Integrating recommendation systems into power material management can assist enterprises in reducing operational costs through precise demand forecasting, inventory optimization, and improved procurement efficiency. Furthermore, this article seeks to explore a cutting-edge recommender system tailored for electrical materials.

The recommendation system focuses on examining user-item interactions, employing methodologies such as matrix decomposition, collaborative filtering, and deep learning to deliver personalized recommendations. These techniques have demonstrated considerable success in the realm of personalized suggestions. Nonetheless, they encounter notable hurdles, such as data sparsity, cold start challenges, and the absence of interpretability in recommendations. In response, an increasing body of research (*Wang et al., 2019a*, *2020*) suggests integrating knowledge graphs into recommendation systems through graph neural networks to leverage their extensive semantic data and better capture user preferences. Since graph neural networks (GNNs) (*Li et al., 2023*) propagate high-order information by layer-wise aggregation from neighboring nodes, this multi-hop information significantly enhances the representation of users. The aggregation mechanism utilized by GNNs successfully overcomes the shortcomings of both embedding-based and path-based approaches, leading to the development of more resilient recommender systems. For instance, *Ji et al. (2023)* proposed a knowledge augmentation model, KMTE, which enhances the efficacy of conversational recommendation systems by integrating time embedding and domain-specific knowledge.

Despite their advantages, GNN-based methods have a significant limitation: they are unable to explicitly capture user intent information, which is crucial in the selection process. For instance, as illustrated in Fig. 1, the traditional GNN approach selects items based solely on user preferences, as seen in the left part of Fig. 1. A user who favors alternating current transformers is more likely to choose alternating current transformers from a range of transformer-related equipment, such as transformer distribution boxes and transformer clamps. However, this approach overlooks the influence of user intent. In fact, users' preference favor the fine-grained embedding of users, while users intent likes the coarse-grained embedding of users. Therefore, the selection behavior that takes intent into account aligns more closely with the user's decision-making process, as depicted in the right part of Fig. 1. In this scenario, the user first intends to purchase electrical transformer equipment and subsequently selects an alternating current transformer based on their preference for that category. User intent can refine the item selection process and provide additional context for recommendations. Additionally, the drawbacks of conventional GNN approaches in capturing user intent arise from their dependence on aggregating

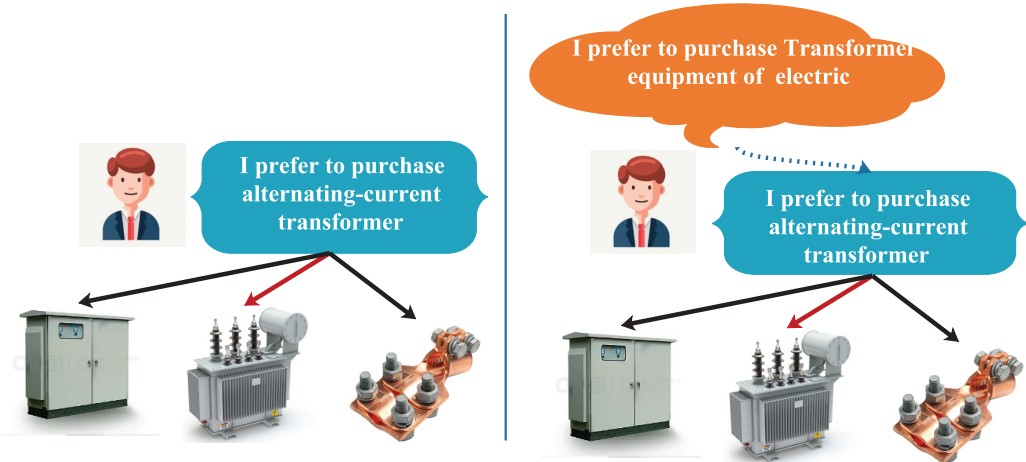

**Figure 1** An illustration demonstrating how intent influences a user's behavior.

items that users have interacted with or related high-level items to construct user representations, which mainly emphasize user preferences. This focus complicates the effective exploration of users' intention data.

To address the aforementioned challenge, we proposes an intent-aware knowledge graph-based model for electrical material recommendation, named IKG-EMR. IKG-EMR models user preferences and intent by utilizing knowledge graphs and user behavior sequences, respectively. In this framework, a graph neural network (GNN) is designed to produce user intent and item embeddings from the tripartite graph of "User-Item-Topic" (the tripartite graph depicts the complex relationships among users, items and topics). Meanwhile, a multi-head Transformer generates user preference embeddings, and a multi-head attention network (Transformer) extracts preferences from user behavior sequences. Ultimately, an adaptive fusion mechanism with an attention network is developed to create a comprehensive user representation by combining user preference and intent features. Additionally, it is important to highlight that the real-life electric dataset applied in this article holds significant potential for enhancing services in electrical power material management, aiding in customer attraction and retention, and ultimately increasing the conversion rates from casual visitors to buyers. In summary, the contributions of this article can be summarized in three key points:

- We construct tripartite graph of "User-Item-Topic" and behavioral sequence, and design a graph neural network and a multi-head Transformer to model user intent embedding and user preference embedding. This approach enables us to obtain comprehensive and high-quality representations of users, thereby enhancing the accuracy of recommendations.
- We design an adaptive fusion with attention network which can effectively incorporate correlation embedding information of users' intent and preference, rather than mere combinations. Unlike traditional attention mechanisms that use scalar weights, our

approach offers enhanced representational capabilities to manage the significance of these two types of user embedding vectors more effectively.

- Extensive experiments on the real-world electrical power dataset demonstrate that IKG-EMR evidently outperforms the state-of-art methods, and further results also validate the effectiveness of IKG-EMR in modeling the intent of users.

# RELATED WORK

The relevant studies on the issue of electrical power material recommendation are summarized from three aspects: session-based recommendations, GNN-based recommendations, and intent-aware recommendations.

## Session-based recommendations

Session-based recommendation focus on predicting the next item for a user based on their past interactions, such as a series of purchases or clicks (*Xu et al., 2019*). Unlike traditional user-item recommendation frameworks, SR emphasizes sequential patterns and utilizes a limited set of user interactions within a brief timeframe, referred to as a session. Early research (*Shani et al., 2005*; *Rendle, Freudenthaler & Schmidt-Thieme, 2010*) commonly employed Markov decision processes (MDPs) to forecast a user's subsequent actions based on their prior behaviors. For instance, *Rendle, Freudenthaler & Schmidt-Thieme (2010)* introduced a model that captures the sequential nature of consecutive clicks, improving the accuracy of predictions for each sequence. Nonetheless, a significant limitation of Markov chain-based models is their assumption that past elements are independent, which can hinder prediction accuracy.

In recent years, a considerable amount of research (*Quadrana et al., 2017*; *Zhao et al., 2020*) has focused on harnessing deep learning techniques, particularly through the development of various recurrent neural networks (RNNs) and attention-based models for session-based recommendations, which have shown promising results. For instance, *Quadrana et al. (2017)* proposed an innovative approach to personalize RNN models by transferring information across sessions and introduced a hierarchical RNN model that refines and propagates the latent hidden states of RNNs throughout user sessions. Additionally, *Li et al. (2017)* employed the embedding from the last-click to represent the user's current interests and built an attention model on this foundation to effectively capture the user's short-term intent. Recently, there has been an increasing interest in graph neural networks (GNNs), which effectively merge node information with topological structures, positioning themselves as an innovative approach within the field of recommender systems (*Wei et al., 2022*; *Zhang et al., 2022*). Several studies (*Wu et al., 2019*; *Zhang et al., 2020*; *Zhu et al., 2023*) have adapted GNNs for session-based recommendation tasks, improving performance by modeling each session as a graph. For instance, the SR-GNN model (*Wu et al., 2019*) employs gated graph neural networks (GGNN) to understand the intricate relationships between item transitions in session-based recommendations. Nonetheless, numerous methods struggle to effectively differentiate the impacts of various historical sessions on the current session. To tackle this problem, *Zhang et al. (2020)* proposed a personalized graph neural network that

incorporates an attention mechanism to better represent how past sessions influence the current one. Despite these advancements, the GNN-based recommendation methods discussed often lack interpretability, making it difficult to assure users of the reliability of the recommended outcomes.

## GNN-based recommendations

In recent years, there has been a rapid rise in the exploration of various GNN variants to tackle a wide range of challenges in graph-related tasks like graph embedding, node classification, and link prediction (*Raza et al., 2024*). These studies have demonstrated significant potential in handling complex graph data (*Gao et al., 2021*; *Wu et al., 2020*; *Ma et al., 2022*; *Luo et al., 2022*). Leveraging the ability of GNNs to process graph-structured data and extract structural information, a mass of GNN-based recommender systems has emerged to address diverse challenges across different recommendation scenarios (*Wang et al., 2021b*, *2021a*). Motivated by the achievements of collaborative filtering (CF) models that utilize neural networks, researchers have delved into graph-based CF techniques (*Berg, Kipf & Welling, 2018*; *Wang et al., 2019d*; *Hu et al., 2021*). For instance, models like graph convolutional matrix completion (GCMC) (*Berg, Kipf & Welling, 2018*) and neural graph collaborative filtering (NGCF) (*Wang et al., 2019d*) apply graph convolution and embedding propagation mechanisms to enhance the representation of users and items. However, these methods often overlook latent relations between users and items. Addressing this limitation, *Hu et al. (2021)* propose a neural graph personalized ranking (NGPR) model to explicitly capture these connections by utilizing the user-item interaction graph along with nonlinear interaction modeling. Although these collaborative filtering methods for graph-structured data provide considerable advantages, they may not fully consider the dynamic evolution of user preferences and consumption motivations, particularly in sequential recommendation scenarios. Models like HOP-Rec (*Yang et al., 2018*) strive to enrich user-item interactions through random walks. The inherent structure of graphs naturally aligns with many data types in recommender systems, such as user-item interactions represented as bipartite graphs and user clickstreams modeled as session graphs (*Ding et al., 2021*).

To tackle the challenges mentioned above, several existing approaches (*Wu et al., 2019*; *Qiu et al., 2019*; *Xu et al., 2019*) have constructed sequential graphs that are generated by the item sequences within the same session. They then employed GNNs to capture the intricate transitions among items, which were deemed complex and challenging for traditional sequential methods (*Ding et al., 2021*). GGNN (*Li et al., 2015*), as a modification of GNNs, integrated GRUs to comprehensively capture internode relations by iteratively integrating influences from adjacent nodes within the graph. Subsequently, SR-GNN (*Wu et al., 2019*) employed GGNN to represent intricate item transition dynamics in session-based recommendation contexts. However, many of these approaches did not adequately distinguish the effects of different historical sessions on the current session. To tackle this challenge, a personalized graph neural network with an attention mechanism (A-PGNN) (*Zhang et al., 2020*) was introduced to address this issue by modeling the influence of historical sessions on the current session. Nevertheless, prior models typically

focused on modeling item-to-item relations within each session, neglecting the opportunity to globally characterize relations across different sessions for enhanced representations of items. To overcome this limitation, *Wang et al. (2021b)* introduced the LP-MRGNN model, which successfully constructed and modeled a multi-relational item graph to extract valuable insights from various sessions and diverse types of user behavior. Motivated by the diverse GNN-based models mentioned above, this study presents a new GNN-based model designed to learn item representations from co-occurrence graphs based on four attributes for session-based recommendations. As far as we know, no previous research has aimed to concurrently model item connections within session graphs alongside item-item transitions in multi-view attribute co-occurrence graphs.

## Intent-aware recommendations

Recently, researchers have strived to embed intent-awareness into recommendation systems in a more direct and unambiguous manner, as noted in the work by *Jannach & Zanker (2024)*. Notably, the results from practical industrial applications underscore the substantial potential of customizing recommendations based on the anticipated intentions of users. For example, in the context of session-based recommendation, the MCPRN model, as presented by *Wang et al. (2019b)*, introduced mixture-channel purpose routing networks. These networks were designed to adaptively learn the diverse purchase intentions of users for each item across different channels (sub-sequences). Meanwhile, *Liu et al. (2020)* put forward a multi-intent translation graph neural network, which aimed to uncover multiple user intents by taking into account the correlations among these intents. Additionally, *Pan et al. (2020)* devised an intent-guided neighbor detector in their ICM-SR model, which was used to retrieve the appropriate neighbor sessions for the representation of neighbors. Distinct from session-based recommendation approaches, another stream of research efforts is centered on modeling the sequential changes in users' interaction behaviors over a more extended time period. *Ma et al. (2020)* proposed the DSSRec model, which featured a seq2seq training strategy. This strategy employed multiple future interactions as a form of supervision and incorporated an intent variable derived from a user's historical and future behavior sequences. The intent variable served to capture the mutual information between an individual user's past and future behavior sequences. It's worth noting that two users with similar intents might have representations that are quite distant in the representation space. In contrast to this approach, in our work, the intent variable is learned from all users' sequences and is utilized to maximize the mutual information among different users who have similar learned intents. Moreover, ASLI (*Tanjim et al., 2020*) utilized a temporal convolutional network along with side information (such as user action types like click, add-to-favorite, *etc*.,) to capture user intent. Subsequently, the learned intents were employed to guide the session recommendation (SR) model in predicting the next item.

Unlike the aforementioned studies, our proposed IKG-EMR models user embedding from both two perspectives by utilizing knowledge graphs and user behavior sequences, respectively, *i.e.*, user preference and user intent. In addition, IKG-EMR designs a GNN to

**Table 1 Notations and corresponding descriptions.**

| Notations | Description |
|---|---|
| $U, V, O, E$ | User set, item set, node set, edge set |
| $S^u$ | Behavior sequence of user $u$, |
| $G = (O, E)$ | Tripartite graph of "User-Item-Topic" |
| $K$ | Number of topics |
| $x_i, x_u$ | Initial embeddings of item $i$ and user $u$ |
| $N_u$ | Number of items connected to user $u$ |
| $r_u^I$ | Intent embedding of user $u$ |
| $r_u^P$ | Preference embedding of user $u$ |
| $r_u, r_i$ | Final embedding of user $u$ and item $i$ |
| $\hat{y}_{(u,i)}$ | The probability of user $u$ purchasing item $i$ |
| $\mathcal{L}(\hat{y})$ | Loss function of IKG-EMR |

produce user intent and item embeddings from the tripartite graph of "User-Item-Topic", which can alleviate the problem of sparsity faced by recommendation systems.

## PRELIMINARY

In this section, we begin by introducing some fundamental definitions and then present the problem formulation. Additionally, Table 1 presents key notations used in this article and their corresponding meanings.

**Definition 1. (Behavioral sequence).** *The behavioral sequence represents a user's click stream, detailing the order of items they have interacted with over time. This sequence captures the items a user has previously purchased. We denote the behavioral sequence of user $u$ as $S^u$.*

Let $U$ and $V$ represent the set of users and items, respectively. For each user $u \in U$, their behavioral sequence $S^u = [v_1, v_2, \cdots, v_N]$ with the ascending order of time can be gotten, where $N = |S^u|$ is the number of purchases of user $u$, $v_i$ denotes the $i$-th item.

**Definition 2. (Item topic generalization).** *For any item i, its title can be generalized through Latent Dirichlet Allocation (LDA) (Blei, Ng & Jordan, 2003) to obtain a probability distribution over topics, denoted as $\theta_i = \{\theta_{i,k}\}, k = 1, 2, \ldots, K, \sum_{(k=1)}^{K} \theta_{i,k} = 1$, where k is the index of the topic and K is the number of topics after item generalization. The topic with the highest probability distribution value in the $\theta_i$ set is selected as the final topic of the item, as given in Eq. (1).*

$$\phi(i) : \phi(i) \rightarrow t_k, 1 \leq k \leq K, \tag{1}$$

*where $t_k$ represents the generalized topic of item $i_m$. Finally, the set of all topics $T = \{t_1, t_2, \ldots, t_K\}$ is obtained, where $t_k = \{i | \forall \phi(i) \rightarrow t_k\}$.*

**Definition 3. (Tripartite graph).** *Let $G = (O, E)$ be the tripartite graph of "User-Item-Topic" as constructed in Fig. 2, where O and E are the sets of nodes and edges in the graph. If the "user-item" purchase matrix $Y_P$ is not empty and there exists a mapping relationship in*

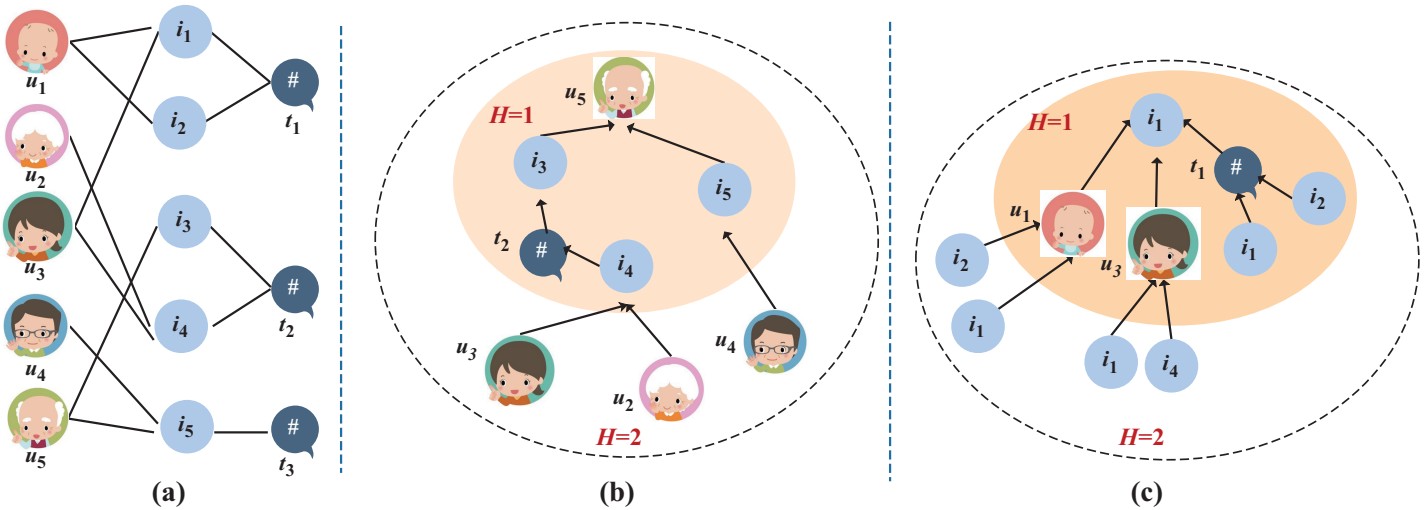

**Figure 2 An example of tripartite graph of "User-Item-Topic" and message propagation.**

the topic set T for "item-topic", edges will be formed between corresponding users and items, as well as between topics and items. The nodes in this graph can be categorized into three classes: the user set $U = \{u_1, u_2, \ldots, u_N\}$, the item set $I = \{v_1, v_2, \ldots, v_M\}$, and the topic set $T = \{t_1, t_2, \ldots, t_K\}$.

Figure 2 illustrates an example of tripartite graph of "User-Item-Topic". Specifically, Fig. 2A is the tripartite graph of "User-Item-Topic" containing five users, five items and three topics. Figures 2B and 2C show the progress of message propagation for $u_5$ and $u_1$ in this tripartite graph, respectively.

**Definition 4. *(Electrical power material recommendation).*** *For a specific user u and their behavioral sequence $S^u$, the objective of the electrical power material recommendation task is to forecast which electrical material $v_i$ this user is most inclined to buy during their next visit.*

## METHODOLOGY

In this section, we outline the IKG-EMR model for recommending electrical materials. We begin by giving an overview of the framework, followed by a detailed explanation of its key components. We also introduce the training process and the loss function utilized in the model.

### Overview

To implement the task of electrical power material recommendation, this article proposes a intent-aware knowledge graph-based mode, named IKG-EMR. As shown in Fig. 3, the input of IKG-EMR contain the tripartite graph of "User-Item-Topic" and the behavior sequence of user. IKG-EMR mainly consists of four modules: user intent embedding and item embedding, user preference embedding, adaptive fusion network, and making recommendation. Specifically, we first use a graph neural network to generate user intent embedding and item embedding from the tripartite graph of "User-Item-Topic", and a

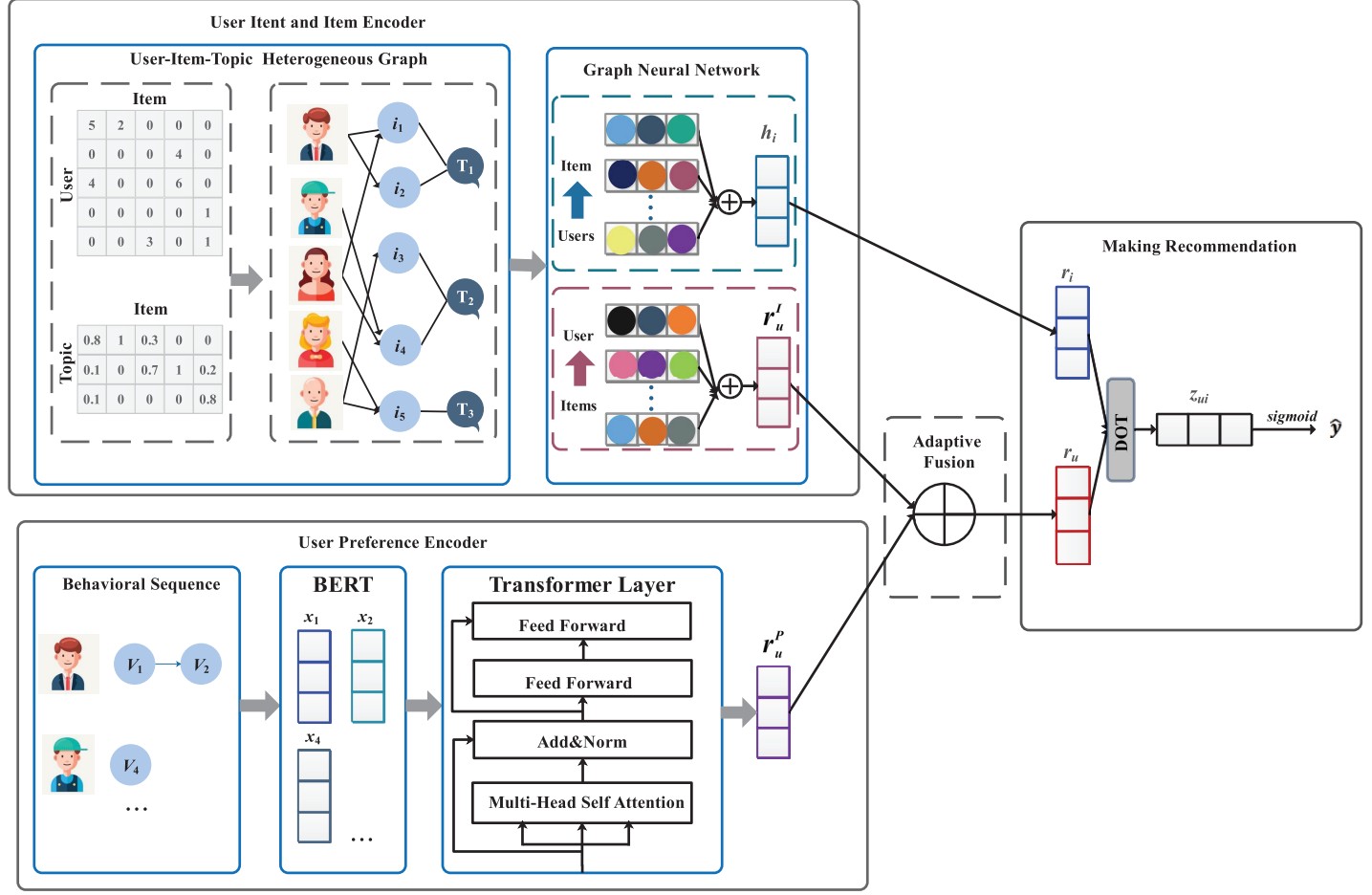

**Figure 3 The overall framework of IKG-EMR.** Generally, IKG-EMR mainly consists of four modules: modeling user intent embedding and item embedding, modeling user preference embedding, adaptive fusion network, making recommendation.

multi-head Transformer to generate user preference embedding, respectively ("Generating User and Item Embedding"). Then, a cohesive representation of users can be achieved by implementing an adaptive gated fusion method, which integrates the intent and preference embeddings of the users (see "Adaptive Fusion Approach"). For recommendation purposes, IKG-EMR computes the purchase probabilities for each item based on the representations of the specified target user and items through interaction modeling. The items with the highest top-$k$ purchase probabilities will constitute the user's recommendation list (refer to "Making recommendation and model training").

## Generating user and item embedding

In this subsection, we detail how to model user's embedding from tripartite graph of "User-Item-Topic" and behavioral sequence of users, respectively.

### Generating user intent embedding

We adopt a GNN information propagation architecture to obtain collaborative filtering signals along the graph structure, thus obtaining representation vectors for both users and items. In practice, items with interaction history often reflect user preferences, while a group of users with interaction records on the same item can be viewed as item features, reflecting similarities between items. Therefore, for users with high-order edge relationships in the tripartite graph of "User-Item-Topic", we conduct embedded propagative learning separately, including information propagation and information aggregation.

**Message propagation.** In a generic single-layer GNN network (*Hu et al., 2021*), for any pair of "user-item" $(u, i)$ with an edge connection in a traditional bipartite graph, the information from item $i$ to user $u$ can be formulated by Eq. (2).

$$m_{u \leftarrow i} = f(x_i, x_u, c_{u,i}), \tag{2}$$

where, $f(\cdot)$ is the encoding function for information, $x_i$ and $x_u$ represent the representation vectors for item $i$ and user $u$, respectively. Specifically, the title of item $i$ is input into a pre-trained bidirectional encoder representations from transformers (BERT) model to obtain vector $x_i$; $x_u$ is obtained through One-Hot encoding of the identifier (ID) for user $u$. $c_{u,i}$ is the attenuation factor used to control the propagation of any edge $(u, i)$, and is represented by the regularization variable $\frac{1}{|N_u|}$. The function $f(\cdot)$ can be implemented by Eq. (3).

$$f(x_i, x_u, c_{u,i}) = \frac{1}{|N_u|} [\alpha W_1 x_i; W_2 x_i], \tag{3}$$

where $N_u$ represents the number of items connected to user $u$, $\alpha = (x_i^T W_3 x_u)$ is designed to measure the information dependency between user $u$ and item $i$ (similar to an attention mechanism), and $W_1$, $W_2$, and $W_3$ are trainable weight matrices in the GNN network used to extract useful information in information propagation. ";" denotes vector concatenation. Ultimately, solving Eq. (3) is equivalent to the form given in Eq. (4).

$$f(x_i, x_u, c_{u,i}) = \frac{1}{|N_u|} [(x_i^T W_3 x_u) W_1 x_i; W_2 x_i]. \tag{4}$$

Similarly, for any pair $(u, i)$ with an edge connection in the tripartite graph of "User-Item-Topic", the propagation from item $i$ to user $u$ is composed of two paths: items directly associated with user $u$, and a set of items belonging to the same latent semantic topic as item $i\{z|z \in \theta(i), z \neq i\}$ that can be represented as in Eq. (5).

$$
\begin{aligned}
m_{u \leftarrow i} &= f(x_i, x_u, c_{u,i}) + \frac{1}{|T(i)|} \sum_{z \in \theta(i), z \neq i} f(x_z, x_u, c_{u,z}) \\
&= \frac{1}{|N_u|} \left( [(x_i^T W_3 x_u) W_1 x_i; W_2 x_i] + \frac{1}{|T(i)|} \sum_{z \in \theta(i), z \neq i} [(x_i^T W_3 x_u) W_1 x_i; W_2 x_i] \right),
\end{aligned}
\tag{5}
$$

where $z$ represents all the items that belong to the same topic as item $i$, $|\theta(i)|$ denotes the number of items included in the topic to which item $i$ belongs, and $W_1$, $W_2$ and $W_3$ are

trainable weight matrices in the GNN network. For example, in Fig. 2B, during the propagation process to obtain the representation vector for user 5, item 3 is a item purchased by user 5. Since it can be inferred from Eq. (1) that items 4 and 3 belong to the same topic $t_2$ ($\theta(3) \rightarrow t_2$), user 5 can receive information about the similar topic item 4 of the purchased item 3 in a single information propagation.

To make the computations in each batch more efficient at this stage, while propagating over the set of other items $\{z \in \theta(i), z \neq i\}$ belonging to the same latent semantic topic $t_k$ as item $i$, we perform random sampling on the item set $\{z\}$. Let $L$ be the maximum number of items sampled from the set that belong to the same topic as item $i$. If $|\{z\}| \leq L$, a random sample is taken within the set $\{z\}$, and then replicated until $|\{z\}| = L$.

**Message aggregation.** Building upon the information propagation, we further aggregate the information propagated from all neighboring nodes of user $u$ (including neighbors in the traditional bipartite graph as well as neighbors obtained through the three-part graph $G$) to obtain the representation vector for user $u$. Specifically, the aggregation function can be defined as Eq. (6).

$$h_u = \sigma\left(\sum_{i \in N_u} m_{u \leftarrow i}\right), \tag{6}$$

where $\sigma(\cdot)$ is the activation function, and in this case, we choose $ReLU(\cdot) = max(0, \cdot)$ as the activation function.

To obtain the final representation vector for user $u$, we transform $h_u$ by using Eq. (7).

$$r_u^I = \sigma(W_u h_u + b_u), \tag{7}$$

where $W_u$ and $b_u$ represent trainable weight matrices and bias vectors, respectively. $r_u^I$ represents the embedding of user $u$ intent learned through embedding propagation in the GNN. Here, we also use ReLU as the activation function.

### Generating item embedding

Similar to the computation embedding of the user $u$ intent $r_u^I$ as shown in "Generating User Intent Embedding", we can also obtain the out of GNN for item $i$ denoted as $h_i$, and the representation vector for item $i$ by aggregating information from users connected to item $i$, denoted as $r_i$. In conclusion, the three-part graph representation learning based on graph neural networks can explicitly utilize connection information to associate user and item representations, and use latent semantic topics as bridges to aggregate more user and item neighbor nodes through the aggregation layer to obtain richer information, thereby obtaining high-quality user and item representation vectors.

The above details the information propagation and aggregation process of a single-layer GNN, where the final representation vector for a item depends solely on its direct neighbors. To capture higher-order relationships between users and items, we extend the GNN from a single layer to multiple layers to propagate embedding information more extensively and deeply. As shown in Fig. 1, the second-order user representation vector can be obtained as follows: firstly, use Eqs. (6) and (7) to aggregate information from neighboring items and topics to obtain the first-order item representation vector and topic

representation vector. Then, once again, aggregate information from first-order items and topics based on neighboring users to obtain the second-order user representation vector. Similarly, as shown in Fig. 1C, second-order item representation vectors can also be obtained.

### Generating user preference embedding

To better capture the evolving preference of user, a multi-head Transformer network is employed for user preference embedding. This network enables the extraction of temporal features from a user's past purchases, enhancing the representation of user features.

For a user $u$ with a behavioral sequence $S^u = [v_1, v_2, \cdots, v_N]$, we encode the elements in the sequence into embeddings $\{v_1, v_2, \cdots, v_N\}$ by using a pre-trained BERT model as given in Eq. (8).

$$E_u = BERT(S_u) = [e_{x_1}, e_{x_2} \ldots, e_{x_n}]. \tag{8}$$

To incorporate the sequential details of the purchased items, we utilize positional embedding operation (*Vaswani et al., 2017*) to update the item embeddings. The positional embedding operation utilizes sine and cosine functions to improve the model's ability to distinguish between positions in the sequence. This can be defined in Eqs. (9) and (10).

$$PE_{(i,2k)} = sin\left(\frac{i}{10{,}000^{\frac{2k}{d}}}\right), \tag{9}$$

$$PE_{(i,2k+1)} = cos\left(\frac{i}{10{,}000^{\frac{2k}{d}}}\right), \tag{10}$$

where $i$ denotes the position of an item in a sequence, $d$ denotes the dimension of the item, $2k$ and $2k + 1$ are the even-numbered and the odd-numbered dimensions, respectively.

After the positional embedding operation, we derive the sequence position embedding representation, denoted as $P_u = \{p_v^1, p_v^2, \ldots, p_v^n\}$. The time series with position information is then obtained by combining $P_u$ with $E_u$, denoted as $Z_u = \{e_{v_1} + p_v^1, \ldots, e_{v_n} + p_v^n\}$.

Next, we utilize the transformer encoder (*Vaswani et al., 2017*) to identify the underlying semantic patterns within the input data. This transformer encoder employs a multi-head self-attention mechanism, featuring several heads, where each head uses dot-product calculations to assess the relationships among the items as given in Eq. (11).

$$\text{SelfAttn}(\boldsymbol{Q}, \boldsymbol{K}, \boldsymbol{V}) = \text{softmax}\left(\frac{\boldsymbol{Q}\boldsymbol{K}_\top}{\sqrt{d_k}}\right)\boldsymbol{V}, \tag{11}$$

where $\mathbf{Q} = Z_u\mathbf{W}_i^Q$, $\mathbf{K} = \mathbf{Z_u}\mathbf{W_i^K}$, and $\mathbf{V} = Z_u\mathbf{W}_i^V$ denote the query, key, and value, respectively, while $\sqrt{d_k}$ serves as a scaling factor to prevent excessively large dot products. For each head $head_i$, the query, key, and value are linearly projected $h$ times, and the results are then concatenated as given in Eqs. (12) and (13).

$$head_i = \text{SelfAttn}(\boldsymbol{Q}, \boldsymbol{K}, \boldsymbol{V}) = \text{SelfAttn}(Z_u\boldsymbol{W}_i^Q, Z_u\boldsymbol{W}_i^K, Z_u\boldsymbol{W}_i^V), \tag{12}$$

$$\text{MultiHead}(\boldsymbol{Q}, \boldsymbol{K}, \boldsymbol{V}) = [head_1; head_2; \cdots; head_h]\boldsymbol{W^O}, \tag{13}$$

where $W_i^Q$, $W_i^K$, $W_i^V$ and $W^O$ are learnable parameters, ";" denotes the concatenation of matrices, and $h$ is the number of heads.

Finally, we apply a feed-forward layer, referred to as FeedForward($\cdot$), which utilizes the ReLU activation function to transform the embeddings and produce the representation of user preferences by Eq. (14).

$$r_u^P = W_u\text{ReLU}(W_s\text{MultiHead}(Q, K, V) + b_s) + b_u, \tag{14}$$

where $r_u^P$ is the embedding of user preference, $W_s$, $W_u$, $b_s$ and $b_u$ are learnable parameters.

## Adaptive fusion approach

Both short-term and long-term components have strengths and weaknesses. It is necessary to accommodate these two components. Instead of using a naive way to combine them, *e.g.*, $\mathbf{r}_u = \mathbf{r}_u^I + \mathbf{r}_u^P$, we devise an adaptive fusion network to evaluate the relevance of user intent and preferences, and aggregate information based on that assessment. Initially, we embed the identifiers (ID) of user $u$ into a representation vector, which serves as the user's profile vector $\mathbf{r}_u^Q$. Given the embedding vectors of user intent $\mathbf{r}_u^I$, user preference $\mathbf{r}_u^P$, and user's profile vector $\mathbf{r}_u^q$ as inputs, the gate vector $\mathbf{F}_u$ is designed to regulate the influence of long-term and short-term preferences, as defined in Eq. (15).

$$\mathbf{F}_u = sigmoid(W_Q r_u^Q + W_I r_u^I + W_P r_u^P + b_u), \tag{15}$$

where $W_q$, $W_s$, $W_l$, $b_u$ are projection parameters.

Finally, the output of preference vector $r_u$ of user $u$ can be calculated by Eq. (16).

$$r_u = (1 - F_u) \odot r_u^I + F_u \odot r_u^P, \tag{16}$$

where $\odot$ denotes element-wise multiplication.

## Making recommendation and model training

The "user-item" interaction modeling layer aims to model the preference level between users and items. Specifically, in the recommendation model framework of IKG-EMR, the interaction score between user $u$'s representation vector $r_u$ and item $i$'s representation vector $r_i$ to predict user $u$'s interaction score with item $i$ as defined in Eq. (18).

$$z_{ui} = \Sigma(r_u^T r_i), \tag{17}$$

where $\Sigma(\cdot)$ is the activation function. Here, we also choose ReLU as the activation function. The final output of the "user-item" interaction modeling layer is the interaction score of user $u$ with item $i$, denoted as $z_{ui}$.

Given the interaction score $z_{ui}$ between user $u$ and item $i$, we use the sigmoid function to obtain the model's output (the probability of user $u$ purchasing item $i$), as given in Eq. (18).

$$\hat{y}(u, i) = sigmoid(z_{ui}), \tag{18}$$

where $sigmoid(\cdot) = \frac{1}{1+e^{-(\cdot)}}$.

**Table 2 Statistics of the electric power materials dataset.**

| Dataset | #Users | #Items | #Interactions |
|---|---|---|---|
| Electric | 80 | 2,195 | 16,598 |

**Note:**
"#" represents the quantity of someone object.

During the model training phase, in terms of recommending items to users, the positive labels are the set of items $i$ that user $u$ actually purchased (interaction exists), denoted as $Y^+$. The negative labels are formed by log-uniform sampling from the item set $I$ excluding the positive labels (interaction does not exist), denoted as $Y^-$. We use binary cross-entropy loss, widely used in recommendation systems, as the loss function for IKM-EMR, as defined in Eq. (19).

$$\mathcal{L}(\hat{y}) = -\sum_{(u,i)\in Y^+\cup Y^-}(y_{u,i}\log(\hat{y}_{u,i}) + (1-y_{u,i})log(1-\hat{y}_{u,i})) + \lambda||\Theta||_2^2, \qquad (19)$$

where $\Theta$ denotes all the learnable parameters in this model, $||\Theta||_2^2$ represents $L2$-regularizer, and $\lambda$ is the regularization coefficient. $y_{u,i}$ represents the probability distribution of user $u$ actually purchasing item $i$. Specifically, if $(u, i) \in Y^+$, then $y_{u,i} = 1$; otherwise, $y_{u,i} = 0$.

We utilize the Adam optimizer (*Zhu et al., 2023*) to minimize the loss function $\mathcal{L}(\hat{y})$ to optimize the parameters in the IKM-EMR model. Compared to existing training schemes, this approach efficiently extracts valuable insights from negative samples, thereby lowering the computational expenses associated with model training.

# EXPERIMENTS

In this section, we start by detailing the experimental setup in "Experimental Setup", which covers the experimental datasets, baseline methods, evaluation metrics, and implementation specifics. Following that, to demonstrate the efficacy of the proposed IKM-EMR, we perform comprehensive comparative experiments against leading baseline methods in "Overall Performance Comparison". Finally, we carry out ablation studies and a hyper-parameter analysis on our model to examine the influence of various components in "Ablation Study". Finally, the impact of hyper-parameters in IKM-EMR on its overall performance is further analyzed in "Hyper-Parameter Analysis".

## Experimental setup

### Datasets

We evaluate the proposed model using a real-world electric power materials dataset provided by Jiangsu Electric Power Materials (http://www.js.sgcc.com.cn), one of China's largest provincial power grid companies. The time interval of this dataset spans from March 18, 2021, to May 9, 2023. The statistics of the preprocessed dataset is summarized in Table 2, and the dataset is available at https://github.com/zgx881205/electric-dataset.

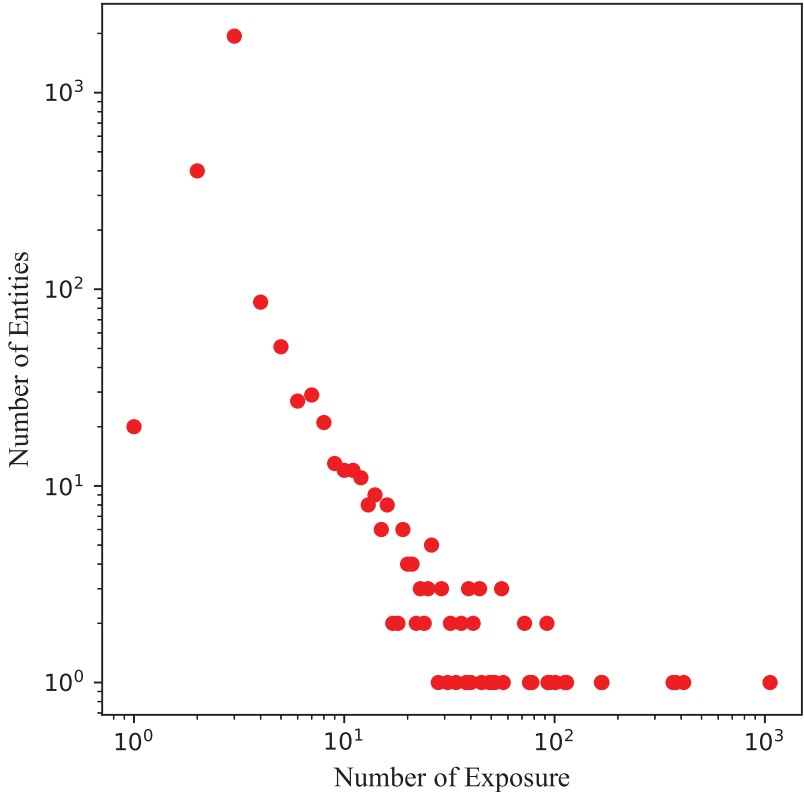

**Figure 4 Long-tail distributions of entities in knowledge graph of electric power materials.**

In practice, knowledge graph of real-world electric power materials dataset faces sparsity and noise, including long-tail entity distributions and irrelevant item-entity links, which limit KGR effectiveness. Using this real-world electrical dataset as an illustration, as depicted in Fig. 4, most entities in knowledge graphs display a long-tail distribution. Accurately modeling semantic transitions in the knowledge graph using Trans-based method necessitates a sufficient number of triplets associated with each entity, posing challenges for precisely capturing relationships between items.

### *Baselines*

We evaluate the proposed IKG-EMR model against two traditional recommendation approaches, namely User-KNN and SVD, two earlier deep learning techniques, *i.e.*, CDL and DeepFM, and also three advanced deep neural graph-based recommendation methods (GCMC, SR-GNN, and KGCN).

- **User-KNN** (*Resnick et al., 1994*). User-KNN utilizes nearest neighbor algorithms to recommend items to users by identifying similar users in the system.
- **SVD** (*Koren, 2008*). SVD employs matrix factorization to decompose the user-item matrix and capture latent factors representing user preferences and item characteristics.
- **CDL** (*Wang, Wang & Yeung, 2015*). CDL integrates deep representation learning for content data with collaborative filtering applied to the feedback matrix, such as rating.

- **DeepFM** (*Guo et al., 2017*). DeepFM integrates matrix factorization techniques with deep neural networks to enhance the recommendation process.
- **KGAT** (*Wang et al., 2019c*). Knowledge graph attention network (KGAT) is a neural architecture that dynamically aggregates node features using attention mechanisms to learn adaptive representations of graph-structured data.
- **CKAN** (*Wang et al., 2020*). CKAN merges collaborative filtering with KG by proposing a heterogeneous propagation method that encodes both approaches.
- **GCMC** (*Berg, Kipf & Welling, 2018*). GCMC utilizes gated graph neural networks (GGNN) to capture transition relationships of items for session-based recommendations.
- **SR-GNN** (*Wu et al., 2019*). SR-GNN employs GGNN to effectively model the intricate transition relationships between items for session-based recommendations.
- **KGCN** (*Wang et al., 2019a*). KGCN integrates knowledge graphs into recommendation systems using graph convolutional networks to improve accuracy by leveraging item relationships and user preferences encoded in the knowledge graph.

### *Evaluation metrics*

In order to assess the performance of IKG-EMR and baseline methods, we utilize evaluation metric Recall@$k$ and Normalized Discounted Cumulative Gain (NDCG@$k$) that are widely used in the related work (*Zhu et al., 2021*; *Liu et al., 2021*).

Recall@$k$ serves as a widely recognized metric for gauging predictive accuracy in recommendation systems. It measures the proportion of accurately recommended items found within the top-$k$ recommendations across all testing scenarios, as defined in Eq. (20).

$$\text{Recall@}k = \frac{1}{|\mathcal{T}|} \sum_{u \in \mathcal{T}} \frac{|G_u \bigcap R_u^k|}{|G_u|}, \tag{20}$$

where $R_u^k$ represents the top-$k$ list of recommendations for each user $u$, while $G_u$ indicates the item purchased by user $u$ in their current session. The test set is denoted by $\mathcal{T}$, and $g_u$ specifies the item purchased by user $u$ in their current session.

Coverage@$k$ refers to the proportion of items in the recommendation results relative to the entire item catalog. A higher proportion indicates that the recommender system covers a broader and more diverse range of products, as defined in Eq. (21).

$$\text{Coverage@}k = \frac{1}{|\mathcal{T}|} \sum_{u,i \in \mathcal{T}} \frac{|R_u^k|}{|i|}. \tag{21}$$

NDCG@$k$ is a widely used ranking metric in session-based recommendation systems, assessing the model's effectiveness in ordering recommended items. Emphasizing the importance of correct item rankings, NDCG@$k$ calculates a score by summing the gains from position $j = 1$ to $j = k$ in the ranking outcomes. This metric reflects the model's ability to prioritize relevant items within the top-$k$ recommendations provided to users, as defined in Eq. (22).

$$\text{NDCG}@k = \frac{1}{|\mathcal{T}|} \sum_{u \in \mathcal{T}} \sum_{j=1}^{k} \frac{2^{Rel(R_u(j)\varepsilon G_u)} - 1}{\log_2^{(1+j)}}, \tag{22}$$

where $Rel(\cdot)$ serves as an indicator function that denotes the reward assigned to the item at position $j$. Specifically, $Rel(\cdot)$ is set to 1, if the item at position $j$ results in a purchase, otherwise $Rel(\cdot)$ is set to 0. The optimal outcome for NDCG@$k$ is 1, signifying perfect alignment between the ground truth ranking and the recommendation list where the desired item is ranked first.

### Implementation details

We conducted experiments for all comparison methods using their optimal parameter configurations. Specifically, cosine similarity was chosen as the similarity metric, with 80 nearest neighbors in Item-KNN and CDL. In DeepFM, the latent variable dimension was set to 10, while all hidden unit sizes were fixed at 256 with dropout probabilities and learning rates at 0.2 and 0.002 in SR-GNN, respectively. The implementation details involve using Surprise (http://surpriselib.com/) for Item-KNN and PMF, while CDL, DeepFM, GCMC, and SR-GNN were implemented using TensorFlow (https://www.tensorflow.org/).

In the IKG-EMR approach, Python's Gensim (https://pypi.org/project/gensim/) library is utilized for topic generalization. The item titles are fed into an LDA model, generating matrices of "word-topic" and "item-topic" probability distributions, along with key high-frequency words associated with each topic. The maximum number of randomly sampled latent semantic topics, denoted as $L$, is set to 5. Despite the strong representation learning abilities of GNNs, precautions against overfitting are necessary. Random sampling is applied to the latent semantic topic set $\theta(i)$, akin to the "item-topic" edge dropout strategy outlined in previous studies. The GNN architecture consists of 2 layers, with a batch size of 64 and a learning rate of 0.001. Notably, leveraging the pre-trained BERT model for unified embedding vectors necessitates setting the default embedding dimension $d$ to 768 for the four attribute types. In the multi-head self-attention mechanism, the number of heads $h$ and dimension of $d_k$ are set 8 and 96, respectively. We use 10-fold cross-validation to provide robust results for different methods. We randomly split the dataset into two parts, respectively, 10% of which as the training set and the rest set as the test set (the class size distribution holds). The IKG-EMR and other neural-based models are designed and trained on a Windows server equipped with a 3.5 GHz Intel I9-11900k CPU and an 24 GB Nvidia GeForce RTX 3090 Ti GPU, using the PyTorch (https://pytorch.org/) framework.

## Overall performance comparison

Table 3 presents a comprehensive comparison of the overall performance between our IKG-EMR approach and the baseline methods. The table is divided into three sections: the first part includes traditional methods, *i.e.*, User-KNN and SVD; the second part comprises neural-based collaborative filtering and matrix factorization methods, *i.e.*, CDL and DeepFM; and the third part showcases GNN-based models, *i.e.*, GCMC, SR-GNN and

**Table 3 Performance comparison of IKG-EMR and baseline methods on the electric dataset.**

| Methods | Top-5 | | | Top-10 | | | Top-20 | | |
|---|---|---|---|---|---|---|---|---|---|
| | Recall@5 | Coverage@5 | NDCG@5 | Recall@10 | Coverage@10 | NDCG@10 | Recall@20 | Coverage@20 | NDCG@20 |
| User-KNN | 0.023 | 0.041 | 0.027 | 0.094 | 0.102 | 0.068 | 0.122 | 0.218 | 0.079 |
| SVD | 0.029 | 0.058 | 0.025 | 0.112 | 0.119 | 0.075 | 0.135 | 0.234 | 0.086 |
| CDL | 0.044 | 0.067 | 0.056 | 0.129 | 0.135 | 0.085 | 0.154 | 0.297 | 0.119 |
| DeepFM | 0.051 | 0.069 | 0.059 | 0.116 | 0.128 | 0.087 | 0.161 | 0.284 | 0.126 |
| KGAT | 0.065 | 0.096 | 0.061 | 0.135 | 0.156 | 0.096 | 0.179 | 0.386 | 0.158 |
| CKAN | 0.070 | 0.093 | 0.067 | 0.142 | 0.168 | 0.114 | 0.203 | 0.397 | 0.177 |
| GCMC | 0.065 | 0.089 | 0.061 | 0.135 | 0.174 | 0.090 | 0.179 | 0.402 | 0.158 |
| SR-GNN | 0.070 | 0.108 | 0.067 | 0.142 | 0.169 | 0.114 | 0.203 | 0.428 | 0.177 |
| KGCN | 0.072 | 0.112 | 0.061 | 0.148 | 0.192 | 0.125 | 0.212 | 0.413 | 0.189 |
| IKG-EMR | **0.083** | **0.124** | **0.068** | **0.162** | **0.242** | **0.134** | **0.223** | **0.489** | **0.201** |

Note:
The highest value for each metric are highlighted in bold.

KGCN. Overall, IKG-EMR consistently outperforms all baselines across the three datasets, as indicated by all evaluation metrics (Recall@$k$, Coverage@$k$ and NDCG@$k$). More specifically, several key observations can be made:

- The neural-based collaborative filtering (CF) and matrix factorization (MF) techniques, such as CDL and DeepFM, demonstrate a marked improvement in performance compared to earlier traditional methods like User-KNN and SVD. One potential reason for this is that neural-based approaches are capable of learning more effective representations of users and items than their traditional counterparts.
- GNN-based methods, such as SR-GNN, which treats behavior sequences as a directed graph-structured data, outperform GAE, which utilizes a bipartite interaction graph (*e.g.*, GCMC), as well as the neural-based CF and MF techniques, *e.g.*, CDL and DeepFM. This suggests that the design of gated GNNs is particularly well-suited for session-based recommendation scenarios. Consequently, we also employ a multi-head Transformer to model user preferences derived from behavioral sequences.
- IKG-EMR consistently outperforms leading personalized GNNs, including SR-GNN, in the context of session-based recommendation tasks, such as SR-GNN, as well as KG-based recommendation systems. This enhancement can be attributed to two main factors: (1) leveraging the strengths of GNNs, IKG-EMR is capable of extracting high-quality representations of user intent; (2) ; and (3) the adaptive fusion mechanism combined with an attention network in IKG-EMR effectively integrates the correlational embedding information of users' intent and preferences.
- IKG-EMR consistently outperforms leading personalized GNNs, including SR-GNN, in the context of session-based recommendation tasks, such as SR-GNN, as well as KG-based recommendation systems. This enhancement can be attributed to three main factors: (1) leveraging the strengths of GNNs, IKG-EMR is capable of extracting high-quality representations of user intent; (2) IKG-EMR utilizes a Transformer-based

**Table 4 Performance comparison of IKG-EMR and three ablation models on the electric dataset.**

| Methods | Top-5 | | | Top-10 | | | Top-20 | | |
|---|---|---|---|---|---|---|---|---|---|
| | Recall@5 | Coverage@5 | NDCG@5 | Recall@10 | Coverage@10 | NDCG@10 | Recall@20 | Coverage@20 | NDCG@20 |
| IKG-EMR (NoP) | 0.056 | 0.087 | 0.045 | 0.128 | 0.187 | 0.105 | 0.172 | 0.399 | 0.151 |
| IKG-EMR (NoI) | 0.062 | 0.091 | 0.052 | 0.139 | 0.195 | 0.118 | 0.192 | 0.412 | 0.168 |
| IKG-EMR (NoA) | 0.071 | 0.104 | 0.054 | 0.154 | 0.215 | 0.114 | 0.213 | 0.454 | 0.184 |
| IKG-EMR | **0.083** | **0.124** | **0.068** | **0.162** | **0.242** | **0.134** | **0.223** | **0.489** | **0.201** |

**Note:**
The highest value for each metric are highlighted in bold.

architecture to capture sequential dependencies in user behavior sequences; and (3) the adaptive fusion mechanism combined with an attention network in IKG-EMR effectively integrates the correlational embedding information of users' intent and preferences.

## Ablation study

Here, we conduct ablation studies to assess the effectiveness of the key components within IKG-EMR by creating three variants of the model: IKG-EMR (NoP), IKG-EMR (NoI) and IKG-EMR (NoA). Specifically, IKG-EMR (NoP) excludes the user preference encoder module, meaning it does not take into account the informativeness of user preferences while learning their representations. IKG-EMR (NoI) omits the user intent encoder from the User Intent and Item Encoder module, thereby disregarding the embedding of user intent in the representation learning process. Finally, IKG-EMR (NoA) removes the adaptive fusion mechanism with the attention network, resulting in a simple combination of the embedding vectors of user intent and preferences without recognizing the differing significance of these two types of embeddings. The results of these ablation studies are listed in Table 4.

First, we compare IKG-EMR with its three variants: IKG-EMR (NoP), IKG-EMR (NoI), and IKG-EMR (NoA). As observed, IKG-EMR exhibits significantly better performance than these three variants, indicating that the user intent encoder, user preference encoder, and the adaptive fusion mechanism with the attention network are critical for the task of electrical power recommendation.

Second, the notable performance decline of IKG-EMR (NoP) demonstrates that modeling user preferences is the most important factor in electrical power recommendation. This is primarily because incorporating user preferences serves as a fine-grained encoder, enabling the recommendation of more items that users actually like, in contrast to the coarse-grained approach provided by the user intent encoder.

## Hyper-parameter analysis

Here, Recall@20, Coverage@20 and NDCG@20 are selected as evaluation metrics, and then we further examine how key parameters, including the number of GNN layers and the message dropout rate, affect IKG-EMR.
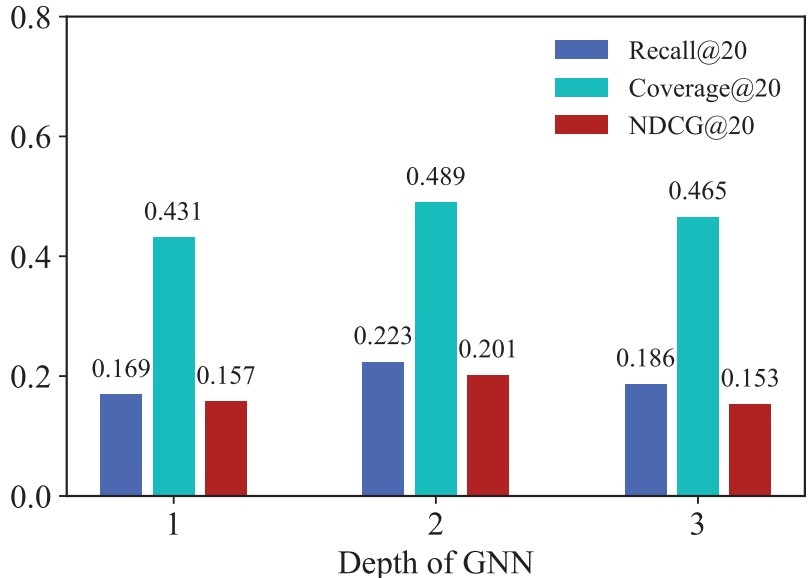

**Figure 5 Results for different numbers of GNN layers.**

### Effect of number of GNN layers

To demonstrate the specific effects of GNNs in IKG-EMR, we explore how varying the number of GNN layers ($d_g$) used for graph representation propagation impacts performance. Figure 5 illustrates the results for GNN layer values ($d_g$) ranging from 1 to 3. Notably, both Recall@20 and NDCG@20 show marked improvement when the GNN depth increases from 1 to 2. A 2-layer GNN achieves the best performance, while the performance of the 1-layer GNN is the lowest. This discrepancy in performance is due to the 1-layer GNN's inability to capture higher-order relationships. Generally, deeper GNNs are anticipated to gather more extensive information from both long-term and short-term behavioral graphs. However, when $d_g$ exceeds 3, the node representations can become less distinguishable, which increases noise within the model. As a result, this can impede further improvements in recommendation performance.

### Effect of dropout

To mitigate overfitting in IKG-EMR and investigate the impact of different dropout rates, the technique of message dropout is utilized here. Figure 6 illustrates the Recall@20 and NDCG@20 results, demonstrating how the message dropout ratio affects performance across two datasets. As indicated, consistent with prior research (*Berg, Kipf & Welling, 2018*), messages are independently dropped out, which enhances the robustness of the embeddings against the variability of individual edges. It is evident that an appropriately chosen dropout ratio allows IKG-EMR to achieve commendable results, while settings that are too low or too high can readily result in underfitting or overfitting issues.

### Effect of learning rate

To investigate how the learning rate affects the performance of IKG-EMR, we tune the learning rate amongst $[10^{-1}, 10^{-2}, 10^{-3}, 10^{-4}, 10^{-5}]$. The corresponding results are

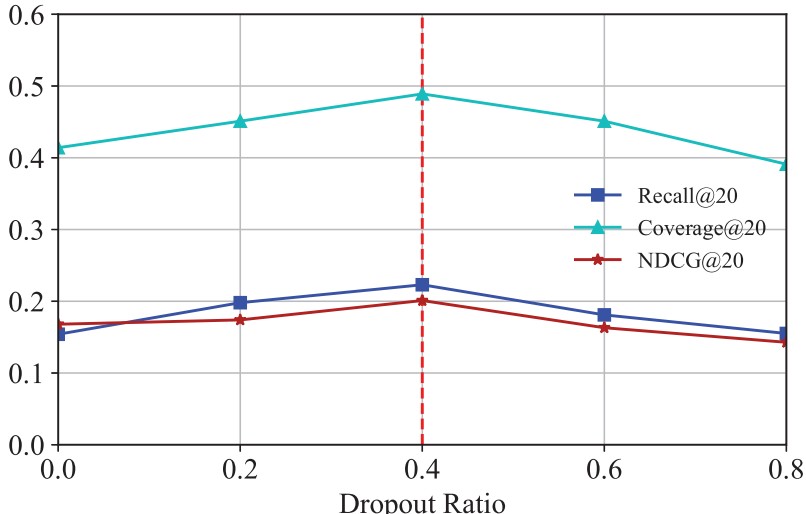

**Figure 6  Results for different dropout of GNN layers.**

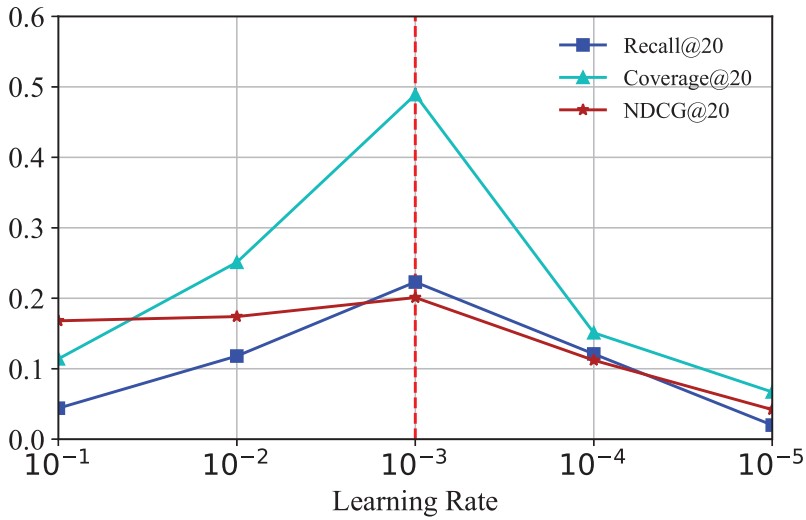

**Figure 7  Results for different learning rate.**

illustrated in Fig. 7. From Fig. 7, we can clearly observe that the performance of IKG-EMR is sensitive to the set of learning rate. The optimal options of learning rate for IKG-EMR is $10^{-3}$.

### Effect of batch size

To investigate how the number of batch size affects the performance of IKG-EMR, we tune the batch size amongst [16, 32, 64, 128, 256]. The corresponding results are illustrated in Fig. 8. It is obvious that when batch size is set to 64, IKG-EMR can achieve the best performance.

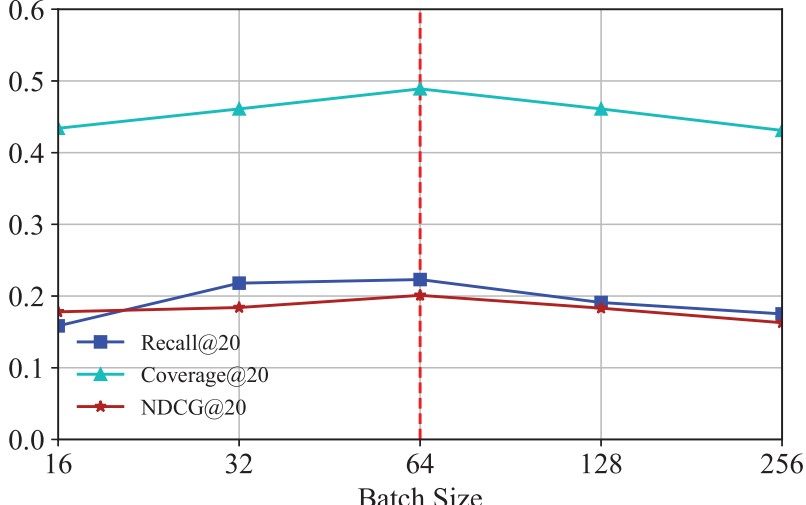

**Figure 8 Results for different batch size.**

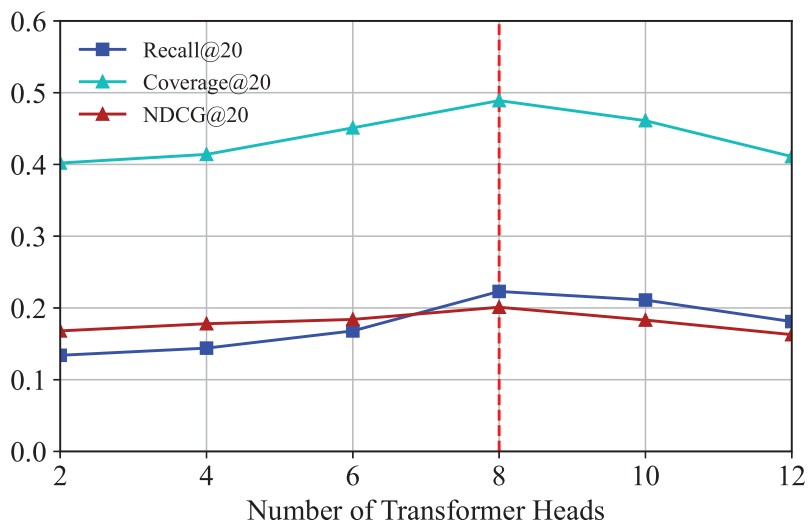

**Figure 9 Results for different numbers of transformer heads.**

### *Effect of transformer heads*

We investigate how the number of transformer heads affects the performance of IKG-EMR. We vary different numbers of transformer heads from 2 to 12 with a step size 2 in the experiment. The corresponding results are illustrated in Fig. 9. As can be seen, the optimal options of the number of transformer heads for IKG-EMR is 8. As the number of heads increases from 8 to 10, the performances descend due to over-parameterization.

## CONCLUSION

This article has addressed the important topic of recommending electrical power materials through the use of GNN. Most of the existing GNN-based recommendation methods still

suffer from the shortcoming of their inability to capture users' intent in recommendations. To tackle this limitation, this article proposes an intent-aware knowledge graph-based model for electrical material recommendation, named IKG-EMR. The major novelties of the IKG-EMR lie in devising a graph neural network to generate user intent embedding from the tripartite graph of "User-Item-Topic", and a a multi-head attention network (Transformer) to generate user preference embedding. Furthermore, an adaptive fusion with attention network is devised to generate high-quality users' representations by integrating user preference and intent features. The experimental results obtained from the real-world electrical dataset demonstrate the advantages of the proposed IKG-EMR model for session-based recommendation tasks.

In practice, IKG-EMR can be deployed for electrical power companies to precisely match materials to project requirements, which can reduce the error of manual screening and improve the efficiency of material procurement. In addition, it can also be extended to other scenarios of product recommendation, such as movies, music and course recommendation. However, IKG-EMR is essentially a black-box deep learning model. Although it achieves high recommendation accuracy, it suffers from a lack of interpretability in its results—meaning its recommendations are difficult for decision-making departments in power companies to understand and trust.

In our future research, we intend to further explore the use of GAT and incorporate the aforementioned information to learn representations of users and products, due to IKG-EMR does not take into account the "number of times a user has purchased a item" (*i.e.*, edge weights) and user demographic information in information dissemination. Additionally, we plan to explore and utilize large language models (LLMs) to further enhance the performance of electrical power material recommendation.

### Funding

This work was supported by the General Project of the Social Science Fund of Jiangsu Province of China 24GLB011, the General Project of Philosophy and Social Science Research of Jiangsu Higher Education Institutions of China 2023SJYB0263 and the University-Level Project of Nanjing University of Finance and Economics XKYC3202411. The funders had no role in study design, data collection and analysis, decision to publish, or preparation of the manuscript.

### Grant Disclosures

The following grant information was disclosed by the authors:
Jiangsu Province of China: 24GLB011.
General Project of Philosophy and Social Science Research of Jiangsu Higher Education Institutions of China: 2023SJYB0263.
University-Level Project of Nanjing University of Finance and Economics: XKYC3202411.

## Competing Interests

Lin Zhao, Ning Luan, Weihua Cheng, Shuming Feng, Hui Wang, and Yongcheng Yang are employed by Jiangsu Electric Power Information Technology Co. Ltd.

## Author Contributions

- Lin Zhao performed the experiments, analyzed the data, performed the computation work, authored or reviewed drafts of the article, and approved the final draft.
- Ning Luan conceived and designed the experiments, performed the computation work, prepared figures and/or tables, and approved the final draft.
- Weihua Cheng performed the computation work, prepared figures and/or tables, and approved the final draft.
- Shuming Feng conceived and designed the experiments, authored or reviewed drafts of the article, and approved the final draft.
- Hui Wang conceived and designed the experiments, prepared figures and/or tables, and approved the final draft.
- Yongcheng Yang performed the computation work, authored or reviewed drafts of the article, and approved the final draft.
- Guixiang Zhu performed the computation work, authored or reviewed drafts of the article, and approved the final draft.

## Data Availability

The data and codes are available in the Supplemental Files.

## Supplemental Information

Supplemental information for this article can be found online at http://dx.doi.org/10.7717/peerj-cs.3023#supplemental-information.

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
