# Peer review of "Intent-aware knowledge graph-based model for electrical power material recommendation"

_PeerJ Computer Science, doi:10.7717/peerj-cs.3023_

## Round 0.1 · original submission · Major Revisions

Dear Authors,

Thank you for submitting your article. Reviewers have now commented on your article and suggest major revisions. We do encourage you to address the concerns and criticisms of the reviewers and resubmit your article once you have updated it accordingly. When submitting the revised version of your article, it will be better to address the following:

1. Many of the equations are part of the related sentences. Attention is needed for correct sentence formation.

2. Equations should be used with correct equation number. Please do not use “as follows”, “given as”, etc. Explanation of the equations should also be checked. All variables should be written in italic as in the equations. Their definitions and boundaries should be defined. Provide proper reference to the governing equations.

3. Some of the variables in the equations listed in the definitions need to be explained. Some mathematical notations are not rigorous enough to correctly understand the contents of the paper. Please recheck all the definition of variables and further clarify these equations.

Best wishes,

Reviewer 1 ·

Basic reporting

Strengths:

-The paper is written in professional English and follows a clear, logical structure
-The introduction effectively establishes the context and motivation for the research
-Figures are relevant and well-labeled, particularly Figure 1, which illustrates the key concept
-Raw data and implementation details are provided, including dataset statistics and code availability

Areas for Improvement:

- The literature review could better differentiate this work from existing intent-aware recommendation systems
-Some technical terms (e.g., "tripartite graph") could benefit from a more detailed explanation when first introduced
-Figure 2 would benefit from a more detailed caption explaining the different components, a, b, and c
-Clarification on User Intent Modeling: While the paper discusses the importance of user intent, it would be beneficial to provide more details on how user intent is explicitly modeled and differentiated from user preferences. A more in-depth discussion on the theoretical underpinnings of user intent in the context of recommendation systems would strengthen the manuscript.

Experimental design

Strengths:

-Comprehensive baseline comparisons including traditional and deep learning approaches
-Detailed ablation studies examining component contributions
- Clear implementation details and hyperparameter specifications

Areas of Improvement:

-Dataset limitations:
*Only 80 users and 2,195 items - relatively small for deep learning
*Need to address potential overfitting concerns
*No cross-dataset validation
-Hyperparameter Sensitivity: The authors have analyzed the impact of the number of GNN layers and dropout rates on the model's performance. However, a more comprehensive hyperparameter sensitivity analysis, including the impact of learning rates, batch sizes, and the number of attention heads in the Transformer, would provide a more complete understanding of the model's robustness.
- Implementation of the multi-head attention mechanism needs more detail on head count and dimension choices

Validity of the findings

Strengths:

-Results demonstrate clear improvement over baseline methods
-Multiple evaluation metrics (Recall@k and NDCG@k) used
-Ablation studies support the contribution of each model component
-Statistical analysis of results appears sound

Areas of Improvement:
-Comparison with More Baselines: The authors have compared IKG-EMR with several baseline methods, including User-KNN, SVD, CDL, DeepFM, GCMC, SR-GNN, and KGCN. However, it would be valuable to include comparisons with other recent GNN-based models that have attempted to address similar challenges, such as Graph Attention Networks (GAT) or other hybrid models that combine GNNs with other deep learning techniques.

-Add time-based split evaluation
-Include diversity metrics
-Analyze model behavior on different user activity levels

-Real-world Applicability: While the experimental results are promising, it would be beneficial to discuss the practical implications of deploying IKG-EMR in real-world electrical power material management systems. For example, how scalable is the model, and what are the potential challenges in integrating it with existing systems?

Reviewer 2 ·

Basic reporting

1. English expression: In line 175, the expression for set of users and set of items is usually ‘set of users’ and ‘set of items’, please check the English expression in the paper again and touch it up.

2. Word misspellings: e.g., in lines 29 and 82, ‘eclectic’ should be ‘electrical’, and in line 216, ‘c(u,i)’ is misspelled. ‘ in line 216 is misspelled, please check it again in full before

Experimental design

1. What is the role of User Behaviour Sequence in IKG-EMR?IKG-EMR ultimately recommends K candidate items, which is the same as the result of implicit recommendation, so what is the difference between User Behaviour Sequence and the implicitly recommended list of user-interacted items? In the comparison experiments, the authors compare IKG-EMR with some implicit recommendation models, while IKG-EMR is a session recommendation model. Is this a fair comparison?

2. The illustration in Figure 3 is not intuitive and lacks the necessary legend explanation, such as the meaning of hi, hu, ri and ru. And Figure 3 is too rough to visualise the process of generating user intent and does not show parts of the methodology, such as the generation process of r^I_u and r_P_u.

Validity of the findings

IKG-EMR uses a pre-trained model, BERT, and the performance of IKG-EMR may be due to the strong representation capability of BERT, if IKG-EMR uses an end-to-end model similar to the comparison model to encode user preferences can it still achieve the optimal results?

Additional comments

The article does not discuss some of the relevant papers.

·

Basic reporting

Yes, the work is clearly understood; good structure and experimentation. I would suggest add one tabular analysis of existing work which can show a comparative study of the existing work and gap of the work.

Experimental design

Yes, good simulation with the data set as well as comparison.

Validity of the findings

Yes, the work was well validated through experiments and datasets.

Reviewer 4 ·

Basic reporting

This is a well-structured and clearly motivated paper that addresses a practical and relatively understudied problem: recommending electrical power materials through an intent-aware framework. The authors propose a thoughtful combination of graph neural networks and Transformer-based behavioral modeling. The integration of user preference and intent, via knowledge graphs and behavior sequence, seems effective, and the adaptive attention fusion module stands out as a strength in bridging the two.
The writing is mostly clear, though often verbose, especially in the methodology section, where explanations could be tightened for clarity. The technical novelty is moderate—the methods themselves are not particularly new—but the application domain adds some uniqueness. However, the main novel angle still feels underdeveloped. Is the contribution primarily a social good impact on the power sector? Or is it a modeling advancement? Clarifying this would help ground the paper’s significance.
The evaluation is sound, with a reasonable set of baselines and metrics. Still, deeper analysis would make the results more compelling. For example, the paper would benefit from interpretability insights or qualitative examples explaining why IKG-EMR performs better. More discussion on the proprietary dataset is also needed—what domain-specific features or biases might exist? How representative is it?
Relevant recent work, such as the study presented in https://arxiv.org/abs/2407.13699, which discusses the limitations and failure modes of GNN-based recommender systems, can be discussed with less verbosity.

A critical reflection on how this paper addresses (or fails to address) such failure modes would strengthen its contribution. Including a table of notations would make the paper easier to follow, and class-wise performance or detailed error analysis could offer more nuance.
Lastly, the question of fairness is absent—are there any implications if certain types of users or items are systematically favored or ignored? Could larger models (like LLMs or foundation models) improve or hinder the goal? Addressing these could open the door to further impact and relevance.

Experimental design

The technical novelty is moderate—the methods themselves are not particularly new—but the application domain adds some uniqueness. However, the main novel angle still feels underdeveloped. Is the contribution primarily a social good impact on the power sector? Or is it a modeling advancement? Clarifying this would help ground the paper’s significance.

The evaluation is sound, with a reasonable set of baselines and metrics. Still, deeper analysis would make the results more compelling. For example, the paper would benefit from interpretability insights or qualitative examples explaining why IKG-EMR performs better. More discussion on the proprietary dataset is also needed—what domain-specific features or biases might exist? How representative is it?

Validity of the findings

A critical reflection on how this paper addresses (or fails to address) such failure modes would strengthen its contribution. Including a table of notations would make the paper easier to follow, and class-wise performance or detailed error analysis could offer more nuance.

Lastly, the question of fairness is absent—are there any implications if certain types of users or items are systematically favored or ignored? Could larger models (like LLMs or foundation models) improve or hinder the goal? Addressing these could open the door to further impact and relevance.

---

## Round 0.2 · accepted · Accept

Dear Authors,

The author is grateful for the attention given to the comments made by the reviewers. It is evident that two of the previous reviewers who requested minor revisions did not accept the invitation to review the revised paper once more. One reviewer who requested a major revision did not submit their review within the expected timeframe. It is the opinion of another reviewer that the submitted paper meets the necessary standards for acceptance. I am also happy with the current version.

Best wishes,

Reviewer 4 ·

Basic reporting

Author have addressed most comment.
I suggest pay special attention to the figures laying out and color scheme as per journal standards.

Experimental design

it is not very novel and some old baselines

Validity of the findings

seems fine